# Muscle synergies analysis shows altered neural strategies in women with patellofemoral pain during walking

Cintia Lopes Ferreira[1], Filipe Oliveira Barroso[2]*, Diego Torricelli[2], José L. Pons[2,3,4,5], Fabiano Politti[1], Paulo Roberto Garcia Lucareli[1]

**1** Department of Rehabilitation Science, Human Motion Analysis Laboratory, Universidade Nove de Julho, São Paulo, Brazil, **2** Neural Rehabilitation Group, Cajal Institute, Spanish National Research Council (CSIC), Madrid, Spain, **3** Legs & Walking AbilityLab, Shirley Ryan AbilityLab, Chicago, IL, United States of America, **4** Department Biomedical Engineering & Dept. Mechanical Engineering, McCormick School of Engineering, Northwestern University, Chicago, IL, United States of America, **5** Department of PM&R, Feinberg School of Medicine, Northwestern University, Chicago, IL, United States of America

* filipe.barroso@cajal.csic.es

**Data Availability Statement:** Data Availability: Data are available on Figshare. Normalized EMG envelopes, as well as muscle synergies components (W, H and V) from each healthy control and patient with patellofemoral pain (PFP)

## Abstract

Several studies suggest that the central nervous system coordinates muscle activation by modulating neural commands directed to groups of muscles combined to form muscle synergies. Individuals with patellofemoral pain (PFP) move differently from asymptomatic individuals. Understanding the neural strategies involved in the execution of tasks such as walking can help comprehend how the movement is planned and better understand this clinical condition. The objective of this study was to compare muscle synergies between women with and without PFP during walking. Eleven women with PFP and thirteen asymptomatic women were assessed using three-dimensional kinematics and electromyography (EMG) while walking at self-selected speed. Kinematics of the trunk, pelvis and lower limbs were analyzed through the Movement Deviation Profile. Muscle synergies were extracted from the EMG signals of eight lower limb muscles collected throughout the whole gait cycle. Kinematic differences between the two groups ($p<0.001$, z-score = 3.06) were more evident during loading response, terminal stance, and pre-swing. PFP group presented a lower number of muscle synergies ($p = 0.037$), and greater variability accounted for ($VAF_{total}$) when using 3 ($p = 0.017$), 4 ($p = 0.004$), and 5 ($p = 0.012$) synergies to reconstruct all EMG signals. The PFP group also presented higher $VAF_{muscle}$ for rectus femoris ($p = 0.048$) and gastrocnemius medialis ($p = 0.019$) when considering 4 synergies. Our results suggest that women with PFP show lower motor complexity and deficit in muscle coordination to execute gait, indicating that gait in PFP is the result of different neural commands compared to asymptomatic women.

## Introduction

Patellofemoral pain (PFP) is a chronic pain condition characterized by a multifactorial origin involving a complex interaction between behavioral, anatomical, biomechanical,

during walking trials at self-selected speed are available at: https://figshare.com/s/88e83948a1685ef8e956.

**Funding:** This work was supported by the European Union's Horizon 2020 research and innovation programme (Project EXTEND—Bidirectional Hyper-Connected Neural System) under Grant 779982 and by the Spanish MCIN/AEI/10.13039/501100011033 and the "European Union NextGenerationEU/PRTR" under Grant agreement IJC2020-044467-I and by The National Council for Scientific and Technological Development – CNPq under Grant 310680/2022-0.

**Competing interests:** The authors have declared that no competing interests exist.

psychological, and social factors [1]. The Movement Deviation Profile (MDP) has shown differences in kinematics [2] and kinetics (such as higher patellofemoral joint stress [1, 3]) in PFP. A reduction in neuromuscular efficiency of the quadriceps muscles [4] has also been reported in individuals with PFP during gait. A recent systematic review found that people with PFP ambulate slower, with lower cadence and a shortened stride length, greater contralateral pelvic drop, and lower knee flexion angles and knee extension moments [5].

Neural control of human locomotion results from the complex interaction between central and peripheral inputs that coordinates the various degrees of freedom of the musculoskeletal system [6]. As a result, similar movements can be produced by different muscle activation and coordination patterns [7]. The hypothesis of muscle synergies suggests that the central nervous system (CNS) does not activate muscles individually but rather sends neural commands that activate specific muscle groups to produce movement [8]. Thus, the analysis of muscle synergies has been used to unravel neural strategies underlying human locomotion [7]. The literature is inconclusive regarding the behavior of muscle synergies in musculoskeletal conditions [9]. Previous studies have shown reduced muscle coordination (*e.g.*, less muscle synergies and higher variability account for each individual muscle) for muscles close to the painful region in women with PFP during lateral step down (LSD) [10], and in individuals with femoroacetabular impingement [11] and gluteal tendinopathy [12] during walking. By identifying a painful episode as a threat, the CNS seems to modify the motor behavior to control pain and protect the painful region [13]. However, if pain persists in addition to the cognitive-behavioral factors, this alteration in motor behavior can contribute to the persistence and chronification of the pain [14].

Several studies have investigated how the CNS controls pathological and normal human locomotion [15–17]. The PFP model is relevant to study how the CNS coordinates muscle activation in musculoskeletal diseases because it provides insight into how the CNS modifies motor behavior in response to pain. Therefore, comprehending the possible neural factors underlying gait execution in individuals with PFP can help understand this clinical condition. We hypothesized that women with PFP exhibit alterations in muscle coordination, based on the literature that shows altered coordination in local muscles in the region where pain is reported [9–11]. We expected to find poor coordination, particularly in the quadriceps muscles, during gait. Therefore, the goal of this study was to compare muscle synergies between women with and without PFP during walking.

## Methods

### Study design

A cross-sectional study was designed, approved by the local ethics committee (protocol number 2.732.037), and conducted at the Núcleo de Apoio à Pesquisa em Análise do Movimento of the Nove de Julho University. All participants agreed to participate and signed the corresponding informed consent.

### Participants

Eleven women with PFP and thirteen asymptomatic women were selected. The assessment stages of this study were similar to our previous study [10].

Inclusion criteria were: age between 18–35 years and body mass index below 30kg/m². For the PFP group (PFPG), subjects should have experienced PFP for at least three months with a minimum Numerical Pain Rating Scale (NPRS) [18] score of three points at least during two of the following activities: squat, run, ascent and descent stairs, kneeling or sitting for prolonged hours with flexed knees.

Exclusion criteria for both groups were: presence of lateral and/or posterior knee pain, history of ligament and/or meniscal injuries, surgical procedures in the lower limbs or spine, ankle sprain, low back pain, more than one episode of patellar dislocation, cardiorespiratory, neurological or musculoskeletal problems that could interfere or prevent the subject from performing the entire assessment and taking controlled medication such as antidepressants. The control group (CG) could not present any lower limb musculoskeletal pain.

## Procedures and instrumentation

Assessments were performed on two non-consecutive days, with a maximum interval of 15 days between them.

On the first day, the candidates for the PFPG were evaluated by two physiotherapists to clinically confirm PFP [19]. All selected subjects underwent 3D kinematics and EMG assessment during gait. Kinematic analysis was performed using an eight-camera Vicon system operating at 240 Hz. According to the Plug-in Gait model, twenty-five retroreflective markers were attached to each subject's skin. EMG signals were captured using an eight-channel wireless acquisition system (EMG System do Brasil Ltda®) composed of bipolar active electrodes with a 1,000 amplification gain and 20–500 Hz bandpass analog filter. EMG signals were recorded with a sampling frequency of 2,400 Hz. Following SENIAM recommendations [20], Ag/AglCl (Miotec®) surface electrodes were positioned on the skin, taking into account the following locations: Adductor Longus (AdLo) [21], Gluteus Medius (GlMe), Vastus Lateralis (VaLa), Rectus Femoris (ReFe), Vastus Medialis (VaMe), Biceps Femoris (BiFe), Tibialis Anterior (TiAn) and Gastrocnemius Medialis (GaMe). The limb with pain or the most painful one for the PFPG and the dominant one for the CG was analyzed. All participants walked at self-selected speed on a 6-meter-long by 1-meter-wide track. Data from at least 25–30 strides were collected, where corresponding EMG signals presented no artifacts or noise that would prevent a trustful analysis.

On the second day, the isometric strength test was performed on hip abductors and lateral rotators, as well as on hip and knee extensors, using a manual dynamometer (Lafayette, IN, USA) [22]. Foot posture index [23] and lunge test [24] were also performed. All subjects answered the questionnaires on the quality of life (SF-36- Medical Outcomes Study 36 –Item Short-Form Health Survey) [25], depression (BECK Depression Inventory) [26], and function (Anterior Knee Pain Scale–AKPS) [18]. The PFPG also answered questionnaires related to the intensity of pain in the last 15 days and during gait assessment (NPRS) [18], symptom duration, catastrophizing (Pain Catastrophizing Scale) [27], and kinesiophobia (Tampa Scale of Kinesiophobia) [28].

## Data analysis

Two researchers with experience in EMG analysis and with no involvement in the data collection process guaranteed, by visual inspection, the quality of EMG data of the eight muscles for 25–30 strides per subject.

The 3D marker trajectories were processed using the Vicon Nexus 2.10 software to estimate the joint centers [29]. Heel strike events from each walking trial were determined by visual inspection after analyzing Vicon data, and were used to define the beginning and the end of each stride [2]. A Woltring filter with a 2 mean square error was applied to reduce the vibratory noise during the marker trajectories due to soft tissue artifacts.

The absolute angles from frontal, sagittal, and transverse planes of the trunk and pelvis segments in relation to the laboratory; frontal, sagittal, and transverse plane of the hip in relation to the pelvis; frontal and sagittal planes of the knee to the thigh; sagittal plane from the foot to the shank; and absolute transverse plane of the foot were analyzed through the Movement

Deviation Profile (MDP) [30]. An MDP curve consisting of normalized data points every 2% of the movement cycle was calculated using the CG data. The average of the 51 points of the MDP curve of all strides was considered for statistical comparison between groups for each gait cycle. MDP was chosen for unifying and simplifying the understanding of kinematic data by combining all analyzed data into a single curve and calculating the deviation of the movement of the PFPG in relation to the CG throughout the movement cycle [30]. MDP was applied in previous studies by our group to compare the kinematics of women with PFP with the kinematics of asymptomatic women in various tasks, including gait, proving to be a suitable method for analyzing possible kinematic differences in this population [2, 10]. In addition, the analysis of kinematics through gait indices is a common practice [31, 32] and among the indices used, the MDP is a more powerful approach than other statistical methods, according to the systematic review presented by Gonçalves SB et al. [33]. The linear parameters: stride and step time, normalized cadence, percentage of first and second double support, single support, stance phase, normalized step length, and speed of each stride were analyzed.

Raw EMG data were exported from the Vicon Nexus software and processed with MATLAB R2018a (The MathWorks, Natick, MA, USA). Concatenated raw EMG signals from each participant were high-pass filtered at 20 Hz, rectified and low-pass filtered at 5 Hz to obtain the EMG envelopes [34]. For each muscle and subject, EMG envelopes were amplitude-normalized by the average of the peaks of each stride and time-normalized by resampling EMG envelopes at each 1% of the cycle [35]. For each subject, normalized EMG envelopes were combined into $m \times t$ ($EMG_o$) matrices, where $m$ is the number of muscles (eight in this study) and $t$ is the time base [number of gait cycles (25–30) x 100] [17]. Muscle synergies components were calculated applying a non-negative matrix factorization (NNMF) algorithm [36]. Mathematically, the algorithm is described in Eq 1:

$$EMGo = WH + e = EMGr + e \qquad (1)$$

Where $W$ is the $m \times n$ matrix specifying the weight of each muscle in each synergy, $n$ is the number of synergies, $H$ is the $n \times t$ matrix specifying the time-varying activation coefficients representing the recruitment of each synergy throughout the cycle. $EMG_r$ is the matrix $m \times t$ resulting from the multiplication of $W$ and $H$ (reconstructed EMG envelopes), and $e$ is the residual error [37]. We considered 3 to 5 synergies ($n$ = 3, 4, 5) as input to the NNMF algorithm. For each $n$, NNMF was run 40 times, and the repetition with the smallest reconstruction error was selected [17].

The minimum number of synergies required to guarantee adequate reconstruction of the EMG signal was determined as the minimum number necessary to obtain variability account for ($VAF_{total}$) $\geq$90%, as well as $VAF_{muscle}$ $\geq$75% for all muscles assessed [34].

CG data were used to obtain two reference matrices: $W_{ref}$ (representing the contribution of each muscle to each synergy) and $H_{ref}$ (containing the activation coefficients and representing how each synergy is modulated over time). For this purpose, $EMG_o$ matrices of all asymptomatic subjects were concatenated, and the NNMF algorithm was applied to obtain $W_{ref}$ and $H_{ref}$ for 3, 4, and 5 synergies. Then, for each PFP subject, synergy vectors (columns of matrices $W$) were ordered based on their similarity with synergy vectors from $W_{ref}$ using normalized scalar product. After ordering the synergy vectors ($W$) and the corresponding activation coefficients ($H$), the grand average of $W$ and $H$ for each group was calculated to compare both groups.

## Statistical analysis

The Shapiro-Wilk test was used to verify the distribution of the analyzed data. When data did not follow a normal distribution, the t-test for independent samples was performed. When

data did not follow a normal distribution, the Mann-Whitney test was used to compare the groups. Data were expressed as a mean and confidence interval (CI 95%). A statistically significant $p$-value was considered when $p \leq 0.05$. Z-score was calculated to quantify the difference in MDP between the two groups [2]. Statistical analysis was performed using IBM SPSS Statistics 25 software (IBM, Armonk, N.Y., U. S. A.).

The statistical parametric mapping (SPM) two-tailed independent t-test was used to compared EMG envelopes of each muscle during all normalized gait cycles between groups. SPM uses a random field theory correction to ensure that only values $\geq$ 5% of the data points of the SPM {t} reach the significance threshold ($\alpha = 0.05$) by chance if the trajectory of the SPM {t} results from an equally smooth random process. Finally, the probability (p) value was calculated for each supra-threshold and infra-threshold region when SPM {t} exceeded the critical threshold [38]. Since data did not follow a normal distribution, SnPM was used for comparisons. All SPM analyses were performed with a spm1d open code (www.spm1d.org).

The Spearman correlation analysis was performed to explore a possible association between $VAF_{total}$ using 4 synergies with pain intensity (NRPS), disability (AKPS), fear of movement (TAMPA), catastrophizing (PCS) and depression (Beck scale) in the evaluated sample of women with PFP. A p-value $\leq$ 0.05 was considered indicative of statistical significance. The correlation coefficients were interpreted as follows: 0.00 to 0.10 = no correlation; 0.10 to 0.39 = weak correlation; 0.40 to 0.69 = moderate correlation; 0.70 to 0.89 = strong correlation; and 0.90 to 1.00 = very strong correlation [39].

## Results

The sample characterization data and the spatiotemporal gait parameters are shown in Tables 1 and 2, respectively. PFPG showed worse scores of AKPS and worse scores of the following domains from SF-36: physical functioning, role physical, bodily pain and vitality. PFPG also walked with higher step length and cadence.

There was a significant difference in the mean (standard deviation) MDP curve between CG 10.4° (1.2°) and PFPG 14.09° (1.4°) (p < 0.001 and z-score = 3.06). MDP analysis showed differences in kinematic behavior between 2% and 12% (loading response), and between 38% and 54% (terminal stance and pre-swing) of the gait cycle, based on 95% confidence interval (Fig 1).

SPM analysis did not find significant differences of individual muscles activation between groups (Fig 2). However, muscle synergies analysis revealed that PFPG presented higher $VAF_{total}$ values when reconstructing EMG signals with 3, 4, and 5 synergies, and higher $VAF_{muscle}$ values for ReFe and GaMe when considering 4 synergies (Table 3).

The minimum number of synergies needed to explain the variability of the EMG signals during gait was different between the two groups (Table 3). In CG, 3 synergies were needed to properly reconstruct the EMG signals in one participant, whereas eight participants needed 4 synergies, and three participants needed 5. In PFPG, three patients needed 3 synergies to reconstruct the EMG signals, and eight patients needed 4 synergies.

According to our criteria ($VAF_{total} \geq 90\%$ and $VAF_{muscle} \geq 75\%$) to determine the minimum number of synergies, we considered 4 muscle synergies to compare synergies components between the two groups (Fig 3). Synergy 1 was active during mid and terminal swing, initial contact, and loading response (H1), and the muscles mainly contributing to this synergy were BiFe and TiAn (W1). Synergy 2, represented by the activity of GlMe, VaLa, ReFe and VaMe (W2), reached its activation peak during the loading response and part of midstance (H2). Synergy 3 had the main contribution of GaMe (W3) and was active during mid and terminal stance, and pre-swing (H3). Initial and mid-swing were controlled by synergy 4 (H4),

**Table 1. Mean and confidence interval (95%) of demographic data, physical assessments and questionnaires scores, and comparison between the groups.**

| | Control Group | PFP Group | |
| --- | --- | --- | --- |
| | mean (CI 95%) | mean (CI 95%) | p-value |
| **Age (years)** | 24.15 (21.14–27.16) | 24.18 (21.33–27.03) | 0.988 |
| **Height (m)*** | 1.61 (1.58–1.64) | 1.66 (1.63–1.69) | 0.011 |
| **Body mass (kg)+** | 56.50 (51.47–61.53) | 64.45 (55.65–73.26) | 0.082 |
| **BMI (kg/m$^2$)** | 21.86 (19.90–23.81) | 23.37 (20.12–26.62) | 0.371 |
| **HABD (%BW)** | 19.62 (15.30–23.93) | 16.38 (12.20–20.57) | 0.254 |
| **HEXT (%BW)** | 21.46 (16.00–26.93) | 19.91 (13.47–26.34) | 0.687 |
| **HLR (%BW)** | 10.16 (8.42–11.90) | 9.94 (8.03–11.84) | 0.850 |
| **KEXT (%BW)** | 34.80 (28.91–40.69) | 32.86 (26.69–39.04) | 0.623 |
| **Lunge test (°)+** | 41.41 (37.18–45.64) | 41.69 (36.94–46.44) | 0.977 |
| **FPI** | 5.08 (3.65–6.50) | 4.27 (2.21–6.33) | 0.476 |
| **BECK (0–63)+** | 6.69 (2.45–10.94) | 6.82 (4.90–8.74) | 0.450 |
| **AKPS (0–100)*+** | 99.69 (99.24–100.15) | 75.27 (69.91–80.63) | 0.000 |
| **PCS (0–52)** | - | 11.18 (7.27–15.10) | - |
| **TAMPA (17–68)** | - | 30.64 (26.87–34.40) | - |
| **NPRS last 15 days (0–10)** | - | 5.18 (4.34–6.02) | - |
| **NPRS during gait (0–10)** | - | 1.91 (0.55–3.27) | - |
| **Symptom duration (months)** | - | 59.09 (20.51–97.67) | - |
| **SF-36 domains:** | | | |
| *Physical functioning (0–100)*+* | 94.23 (88.08–100.38) | 74.55 (63.36–85.73) | 0.000 |
| *Role physical (0–100)*+* | 98.08 (93.89–102.27) | 77.45 (63.50–91.41) | 0.001 |
| *Bodily pain (0–100)*+* | 83.85 (77.75–89.94) | 66.09 (56.87–75.31) | 0.002 |
| *General health (0–100)* | 79.38 (68.07–90.70) | 70.27 (58.03–82.51) | 0.242 |
| *Vitality (0–100)** | 61.92 (49.43–74.41) | 41.00 (29.10–52.90) | 0.015 |
| *Social functioning (0–100)+* | 81.73 (66.12–97.34) | 70.00 (54.42–85.57) | 0.108 |
| *Role emotional (0–100)+* | 74.36 (50.88–97.84) | 60.00 (33.88–86.11) | 0.276 |
| *Mental health (0–100)* | 73.23 (63.15–83.31) | 60.00 (49.66–70.34) | 0.057 |

*: statistically relevant difference p ≤ 0.05

+: Non-parametric data, Mann-Whitney test was used; CI: confidence interval; m: meters; kg: kilogram; BMI: body mass index; kg/m$^2$: kilograms per meter square; %BW: body weight percentage; HADB: hip abductors; HEXT: hip extensors; HLR: lateral hip rotators; KEXT: knee extensors; FPI: foot posture index; BECK: Beck depression inventory; AKPS: anterior knee pain scale; SF-36: 36-item short-form health survey questionnaire; PCS: pain catastrophizing scale; TAMPA: Tampa scale of kinesiophobia; NPRS: numerical pain rating scale.

composed by AdLo and TiAn (W4). There were no differences between groups in terms of recruitment of each synergy throughout the gait cycle (Hs) and the weight of each muscle in each synergy (Ws) (Fig 3).

We found no significant correlation between VAF$_{total}$ and clinical variables of pain intensity (NRPS), disability (AKPS), fear of movement (TAMPA), catastrophizing (PCS) and depression (Beck scale) in the evaluated sample of women with PFP (p > 0.05, see Table 4).

## Discussion

This study compared the coordination of lower limb muscles in women with and without PFP while walking at self-selected speed. Our results add new evidence that women with PFP present reduced motor complexity and muscle coordination during gait (PFPG presented a lower

**Table 2. Mean and confidence interval (95%) of spatiotemporal gait parameters and comparison between the groups.**

|  | Control Group | PFP Group |  |
|---|---|---|---|
|  | mean (CI 95%) | mean (CI 95%) | p-value |
| Stride Time (s) | 1.13 (1.08–1.17) | 1.08 (1.05–1.12) | 0.100 |
| Step Time (s) | 0.56 (0.54–0.58) | 0.54 (0.52–0.55) | 0.090 |
| Normalized Cadence (stride/s) * | 0.88 (0.85–0.92) | 0.93 (0.91–0.96) | 0.033 |
| 1st Double Support (% gait cycle) | 11.92 (10.92–12.92) | 11.18 (10.20–12.16) | 0.262 |
| Single Support (% gait cycle) | 38.36 (37.62–39.10) | 38.96 (38.04–39.88) | 0.271 |
| 2nd Double Support (% gait cycle) | 11.61 (10.81–12.41) | 10.95 (9.92–11.98) | 0.267 |
| Stance Phase (% gait cycle) | 61.90 (60.82–62.98) | 61.09 (59.99–62.19) | 0.264 |
| Normalized Step Length (m) * | 0.61 (0.58–0.63) | 0.68 (0.66–0.71) | 0.000 |
| Normalized Speed (m/s) | 0.56 (0.54–0.58) | 0.55 (0.53–0.56) | 0.336 |

*: statistically relevant difference $p \leq 0.05$; CI: confidence interval; s: second; m: meters; m/s: meters per second; Normalized Cadence = cadence × sqrt (body height/mean body height); Normalized Step Length = step length/(body height/mean body height); Normalized Speed = speed/sqrt (body height/mean body height).

number of muscle synergies, higher $VAF_{total}$ and $VAF_{muscle}$ for ReFe and GaMe) in comparison with women without PFP. This confirms our hypothesis that women with PFP exhibit alterations in muscle coordination.

A lower number of synergies indicates less complexity in terms of muscle coordination (*i.e.*, a smaller number of independently activated synergies may hinder the proper execution of the same biomechanical subtasks of walking [34]) and can be associated with increased co-activation levels between muscles and with coupling and fusion mechanisms of synergies in patients with neural injuries [17, 40]. In this study, PFPG presented a lower number of synergies during gait. Similar results were observed in individuals with stroke [15, 34], spinal cord injury [17] and cerebral palsy [16]. Nevertheless, most of the studies on musculoskeletal disorders do not usually report differences in the number of synergies during gait [9]. Results also showed higher $VAF_{total}$ in PFPG, which is in line with other findings in tasks like LSD in PFP [10] and gait in patients with femoroacetabular impact [11], gluteal tendinopathy [12] and experimentally induced pain in the low back and calf [41]. The presence of pain on the assessment day and the duration of symptoms reported by the PFPG (59.09 months) can partly justify the results obtained. In painful situations, the CNS seems to redistribute activity intra and between the muscles and generate adaptations in motor variability to protect the musculoskeletal system from movements associated with a painful experience [42].

Different tasks may demand different complexity in motor coordination, which can be assessed by the number of synergies extracted [43]. While we observed differences in the number of synergies between PFPG and CG during gait, the same was not observed in LSD task [10]. We can infer that due to frequent execution of gait as an activity of daily living, the CNS has more opportunities to readjust the musculoskeletal system and make neuromuscular adaptations than in other movements less frequently executed.

ReFe presented poorer muscle coordination (higher $VAF_{muscle}$). The ReFe composed synergy 2 along with GlMe, VaLa and VaMe, which was mainly active in the loading response, and kinematic differences were found at this moment in the gait cycle (Fig 1). Literature has shown that local muscles at regions where the pain is reported present reduced muscle coordination [10–12]. Furthermore, individuals with PFP present local hyperalgesia in the knee, despite not being related to the knee's muscle strength and angular kinematics [44], which may also contribute to a different neuromuscular behavior in these subjects. Individuals with

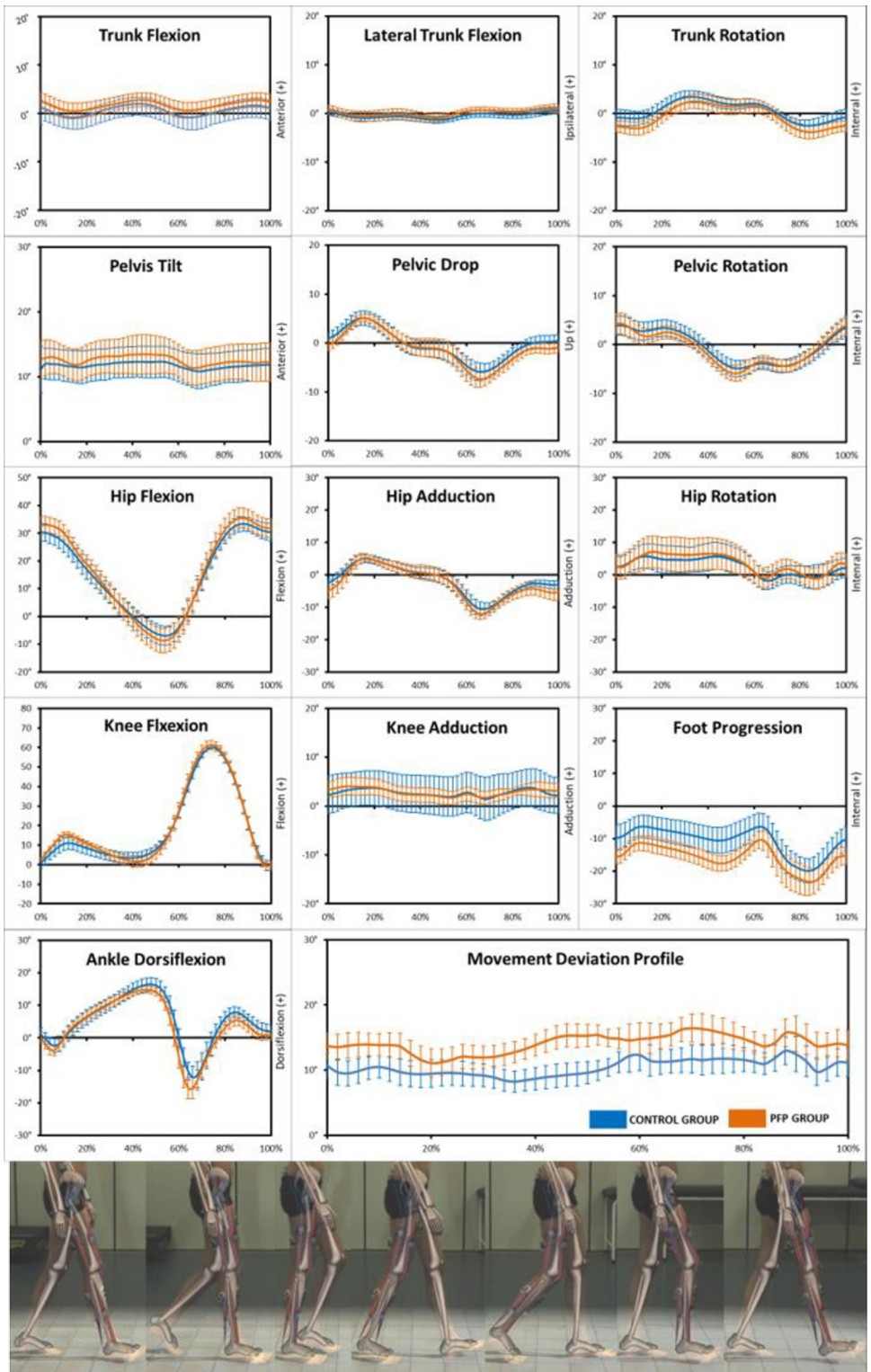

**Fig 1. Kinematics data from sagittal, frontal, and transverse planes of the trunk, pelvis, hip, knee, and ankle during the entire gait cycle.** Movement Deviation Profile (MDP) chart (mean and 95% confidence intervals bands) summarizes the 13 angles curves of kinematics data analyzed for the control group (blue) and PFP group (orange) during gait.

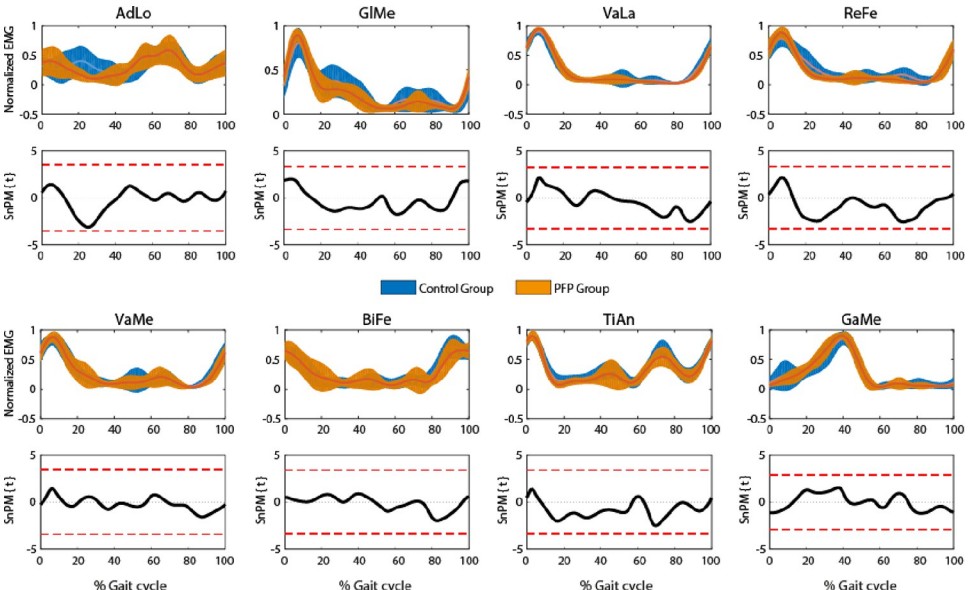

**Fig 2. EMG envelopes from the muscles assessed in this study (for the entire gait cycle) are represented in the first and third rows, for the control group (blue) and PFP group (orange).** SPM analyzes are represented in the second and fourth rows. AdLo: adductor longus; GlMe: gluteus medius; VaLa: vastus lateralis; ReFe: rectus femoris; VaMe: vastus medialis; BiFe: biceps femoris; TiAn: tibialis anterior; GaMe: gastrocnemius medialis.

PFP also present less representation of the quadriceps in the primary motor cortex, indicating impairment in the intermuscular control and coordination, and the adoption of more simplified movement strategies for these patients [45]. Finally, it was observed in animal models that

**Table 3. Mean and confidence of interval (95%) of number of synergies, VAF$_{total}$ considering 3, 4, and 5 synergies and VAF$_{muscle}$ for 4 synergies and comparison between groups.**

|  | Control Group | PFP Group |  | Effect |
| --- | --- | --- | --- | --- |
|  | mean (CI 95%) | mean (CI 95%) | p | Size |
| **Number of synergies*** | 4.23 (3.87–4.59) | 3.73 (3.41–4.04) | 0.037 | 0.19 |
| **VAF$_{total}$ using 3 synergies*** | 0.85 (0.84–0.87) | 0.88 (0.86–0.90) | 0.017 | 1.50 |
| **VAF$_{total}$ using 4 synergies*** | 0.91 (0.90–0.92) | 0.93 (0.92–0.94) | 0.004 | 1.26 |
| **VAF$_{total}$ using 5 synergies*** | 0.95 (0.94–0.96) | 0.96 (0.96–0.97) | 0.012 | 1.00 |
| **VAF$_{muscle}$ using 4 synergies** |  |  |  |  |
| **AdLo+** | 0.94 (0.90–0.98) | 0.97 (0.95–1.00) | 0.287 | - |
| **GlMe** | 0.88 (0.85–0.91) | 0.89 (0.86–0.93) | 0.611 | - |
| **VaLa+** | 0.92 (0.89–0.94) | 0.93 (0.91–0.96) | 0.121 | - |
| **ReFe*** | 0.89 (0.86–0.92) | 0.93 (0.91–0.95) | 0.048 | 0.97 |
| **VaMe** | 0.90 (0.89–0.92) | 0.92 (0.89–0.95) | 0.361 | - |
| **BiFe+** | 0.91 (0.86–0.96) | 0.93 (0.90–0.96) | 0.600 | - |
| **TiAn** | 0.88 (0.85–0.92) | 0.88 (0.84–0.92) | 0.879 | - |
| **GaMe+*** | 0.94 (0.89–0.99) | 0.99 (0.97–1.00) | 0.019 | 0.86 |

*: statistically relevant difference p ≤ 0.05

+Mann-Whitney test; CI: confidence interval; AdLo: adductor longus; GlMe: gluteus medius; VaLa: vastus lateralis; ReFe: rectus femoris; VaMe: vastus medialis; BiFe: biceps femoris; TiAn: tibialis anterior; GaMe: gastrocnemius medialis. The effect size was calculated using Cohen's d index considering 0–0.2 = no effect, 0.2–0.5 = small effect, 0.5–0.8 = intermediate effect and effect greater than 0.8 = large effect.

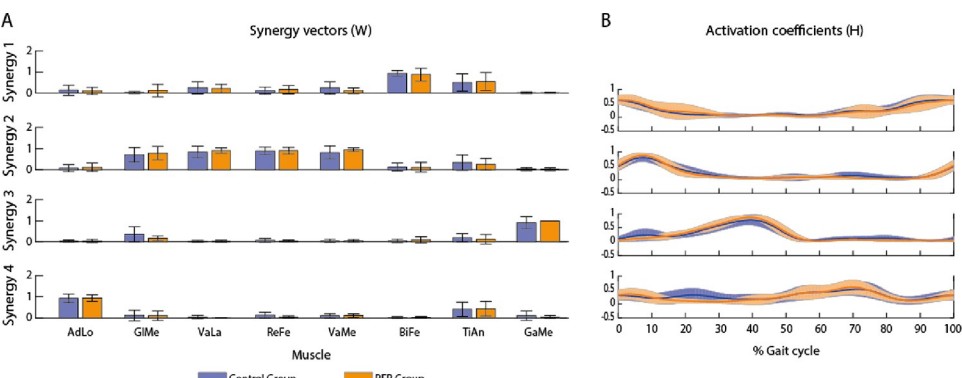

**Fig 3.** The weight of each muscle in each synergy (Ws) (A) and the recruitment of each synergy throughout the gait cycle (Hs) (B) during the gait cycle, considering 4 synergies. AdLo: adductor longus; GlMe: gluteus medius; VaLa: vastus lateralis; ReFe: rectus femoris; VaMe: vastus medialis; BiFe: biceps femoris; TiAn: tibialis anterior; GaMe: gastrocnemius medialis.

the CNS reorganizes muscle activity of the quadriceps to avoid excessive stress and joint tension at the knee joint [46, 47].

PFPG also presented poorer muscle coordination of the GaMe (higher $VAF_{muscle}$), which was the main muscle composing synergy 3 and responsible for double support (Fig 3). GaMe is a biarticular muscle that assists in controlling the ankle in the single support and propulsion in second double support [48], and is also a knee flexor. GaMe can be influenced by neuromechanical adaptations that individuals with PFP present after experiencing pain.

In addition to the analysis of muscle synergies, differences in individual EMG patterns have been found in some neurological diseases such as stroke [49, 50] and cerebral palsy [50–52], for example, but also in PFP [22, 53, 54]. However, these studies mostly focused on hip and knee muscles and compared differences in the activation onset time between pairs of muscles. In our study, we used SPM analysis to look for individual differences in muscle activation between PFPG and CG, but did not find significant differences (Fig 2). In this regard, it is important to highlight that while SPM compares the mean of the linear envelopes of the EMG signals from each subject of each group, the analysis of muscles synergies considers the stride-by-stride variability of each subject and group. Moreover, although we did find differences in the number of muscle synergies between the two groups, no differences were found in terms of recruitment of each synergy throughout the gait cycle (Hs) and the weight of each muscle in each synergy (Ws).

Differences in kinematic behavior in the PFPG can be explained by changes in neural control rather than by clinical variables, as both groups had similar muscle strength, dorsiflexion range of motion (lunge test), and foot posture. The CNS seems to adopt different solutions to protect the painful region, and the differences in the $VAF_{total}$ and $VAF_{muscle}$ indicate impairment in the neural control of the movement in painful conditions [41].

**Table 4. Correlation between $VAF_{total}$ using 4 synergies and anterior knee pain scale (AKPS), pain catastrophizing scale (PCS), numerical pain rating scale (NPRS), Tampa scale of kinesiophobia (TAMPA) and Beck depression inventory (BECK).**

|  |  | AKPS | PCS | NPRS | TAMPA | BECK | VAFtotal |
|---|---|---|---|---|---|---|---|
| **$VAF_{total}$** | Spearman's rho | -0.03 | 0.49 | 0.27 | 0.27 | 0.18 | - |
|  | p-value | 0.94 | 0.13 | 0.42 | 0.42 | 0.58 | - |
|  | Upper 95% CI | 0.58 | 0.84 | 0.74 | 0.75 | 0.71 | - |
|  | Lower 95% CI | -0.61 | -0.15 | -0.39 | -0.39 | -0.46 | - |

*: statistically relevant difference $p \leq 0.05$

It is plausible that clinical and psychosocial variables such as catastrophizing, fear of movement and pain would lead individuals with PFP to use different neural strategies to protect against pain during movement. Studies showed an association between kinesiophobia and the reduction of cadence and peak of knee flexion during stair descent [55], as well as with peak of internal hip rotation increase during single-leg drop vertical jump [56]. In a parallel analysis, which was not in the scope of this study, we found no association between $VAF_{total}$ and clinical variables of pain intensity (NRPS), disability (AKPS), fear of movement (TAMPA), catastrophizing (PCS) and depression (Beck scale). In individuals who suffered an incomplete spinal cord injury, no association was also observed between the number of synergies and the functional scales used to evaluate these patients, indicating that neural strategies for movement performance do not necessarily relate to clinical variables [17]. Individuals with PFP present high levels of catastrophizing, fear of movement, depression, and anxiety and these variables are associated with pain and disability [57]. However, catastrophizing and kinesiophobia are not associated with objective function measured in tests such as forward step-down, single leg hop test, and modified star balance test [58]. In this regard, it is important to highlight that the relationship between biomechanical and clinical variables is still unclear. Therefore, there is a need for future studies with a larger sample size that may better clarify the relationship between neural strategies and clinical and psychosocial variables, which can help achieve better clinical management of patients with PFP.

The present study has also some limitations. The number of assessed muscles does not represent all the muscles activated to execute gait, and this can directly affect the calculation of the number of muscle synergies to describe the task. However, we selected the muscles that exert a primary function in the three main joints of the lower limbs, providing an overall panorama of the task. Our results do not allow us to infer the cause or effect of PFP but provide information on how these patients move and the possible neural factors involved in this condition, which are not possible to identify in the conventional and individual analysis of the EMG signal. The sample size was small, which again begs the question regarding generalizability. In fact, the reduced small sample size may have affected the results on individual muscle activation differences between groups, which were analyzed with SPM. Although qualitative differences between groups can be observed in the activation of some muscles (Fig 2), these did not reach the significance level of 0.05. In the long term, prospective studies investigating the influence of neural factors in the appearance and permanence of the PFP, as well as clinical trials approaching these factors in the treatment plan of these patients, can better elucidate the importance of the study of these muscle synergies in this group of patients. Finally, this study was carried out only with women volunteers, as women are twice as likely as men are to develop patellofemoral pain [59]. Although muscle synergies are similar in younger adults between females and males, this similarity is lower in older females as compared to older males [60]. Therefore, this study should be replicated in men volunteers presenting PFP to assess differences in motor control during gait compared to men with no PFP.

## Conclusion

Women with PFP present lower motor complexity in terms of muscle coordination during walking, indicating that gait in PFP is the result of different neural strategies compared to asymptomatic women.

## Acknowledgments

The authors would like to thank the Universidade Nove de Julho (UNINOVE) for providing the evaluation facilities used in the present study.

## Author Contributions

**Conceptualization:** Cintia Lopes Ferreira, Filipe Oliveira Barroso, Diego Torricelli, Paulo Roberto Garcia Lucareli.

**Data curation:** Cintia Lopes Ferreira, Paulo Roberto Garcia Lucareli.

**Formal analysis:** Cintia Lopes Ferreira, Filipe Oliveira Barroso, Paulo Roberto Garcia Lucareli.

**Investigation:** Cintia Lopes Ferreira, Filipe Oliveira Barroso.

**Methodology:** Cintia Lopes Ferreira, Filipe Oliveira Barroso, Diego Torricelli, José L. Pons, Fabiano Politti, Paulo Roberto Garcia Lucareli.

**Project administration:** Paulo Roberto Garcia Lucareli.

**Software:** Filipe Oliveira Barroso.

**Supervision:** Paulo Roberto Garcia Lucareli.

**Writing – original draft:** Cintia Lopes Ferreira, Filipe Oliveira Barroso.

**Writing – review & editing:** Cintia Lopes Ferreira, Filipe Oliveira Barroso, Diego Torricelli, José L. Pons, Fabiano Politti, Paulo Roberto Garcia Lucareli.

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
