## [Decision Letter · Decision Letter 0]

6 Apr 2023

PONE-D-23-03138MUSCLE SYNERGIES ANALYSIS SHOWS ALTERED NEURAL COMMANDS IN WOMEN WITH PATELLOFEMORAL PAIN DURING WALKINGPLOS ONE

Dear Dr. Barroso,

Thank you for submitting your manuscript to PLOS ONE. After careful consideration, we feel that it has merit but does not fully meet PLOS ONE’s publication criteria as it currently stands. Therefore, we invite you to submit a revised version of the manuscript that addresses the points raised during the review process.

We look forward to receiving your revised manuscript.

Kind regards,

Yaodong Gu

Academic Editor

PLOS ONE

Additional Editor Comments:

The methods part shall be more detail, especially interpretation of coordination.

Reviewers' comments:

Reviewer's Responses to Questions

**Comments to the Author**

1. Is the manuscript technically sound, and do the data support the conclusions?

Reviewer #1: Partly

Reviewer #2: Yes

2. Has the statistical analysis been performed appropriately and rigorously? 

Reviewer #1: Yes

Reviewer #2: Yes

3. Have the authors made all data underlying the findings in their manuscript fully available?

Reviewer #1: No

Reviewer #2: Yes

4. Is the manuscript presented in an intelligible fashion and written in standard English?

Reviewer #1: No

Reviewer #2: Yes

5. Review Comments to the Author

Reviewer #1: General commentaries:

The paper is promising, but certain issues require clarification.

My primary concern is how the complexity of neuromuscular control can differ between groups without changes in individual muscle patterns. Additionally, a more comprehensive interpretation of coordination is necessary, particularly concerning the term "muscle coordination of quadriceps" and the physiological significance of the VAF muscle.

Specific commentaries:

Abstract:

Line 54. What joint showed differences in kinematics?

Introduction:

The lack of information is not always a good argument to research. Why is the PFP model relevant to study how the CNS organizes (coordinates) muscles in musculoskeletal diseases?

Hypothesis:

The hypothesis lacks clarity. How does muscle synergy aid in comprehending the influence of a single muscle's coordination? For this reason, muscle synergy analysis may not be necessary to assess quadriceps coordination. Instead, intermuscular coordination based on time correlation could be a better alternative for this hypothesis. Please clarify.

Method.

Line: 175. Did you check if they are outliers of EMG peaks? Some authors used the median of the peaks, and others the maximum value. Please clarify

Line 213. Did you check the normality of data using SPM? In the case of having a non-normal distribution, you should use the non-parametric tools of SPM. In addition, the 5% as a threshold is arbitrary. Can you give the technical reason for using this threshold?

Results:

Did you check the association between clinical measures and muscle synergy analysis, for instance, VAF values?

Discussion:

Line 314. How can a decrease in muscle complexity be explained without individual muscle changes?

Line 315. But muscle coordination at what level?

Reviewer #2: This study investigated muscle activation and lower kinematics differences between PFP and normal counterparts. The results revealed that women with PFP show lower motor complexity and deficit in muscle coordination to execute gait, indicating that gait in PFP gait is the result of different neural commands compared to asymptomatic women.

The title is not a direct connection between muscle synergy and neural commands. Besides, this study doesn’t prove neural commands changed. The same issue exists in the lines 45-47.

Lines 65-68, a more comprehensive background regarding reasons causing PFP, treatment procedure, and biomechanics change is expected.

the reason why only female subjects were selected is not explained in the manuscript. I would not feel stranger for adding this as one of the limitations.

Lines 134-137, The authors took space to introduce experimental settings, but the test protocol was too ambiguous.

Lines 154-155, please provide more details for identifying gait spatiotemporal parameters.

In Table 1, SF-36, some indexes were confusing. Adding related content in the text.

6. PLOS authors have the option to publish the peer review history of their article (what does this mean?). If published, this will include your full peer review and any attached files.

Reviewer #1: No

Reviewer #2: No

---

## [Author Response · Author response to Decision Letter 0]

20 Jun 2023

Comments from Editor and reviewers:

Editor Comments:

The methods part shall be more detail, especially interpretation of coordination.

Authors: The manuscript has been revised to address the comments addressed by the reviewers (including the interpretation of coordination) so that this revised version can fully meet PLOS ONE’s publication criteria. Importantly, we did a slight change in the title (“neural strategies” instead of “neural commands”) to address one of questions raised by one reviewer, so the new title now reads “Muscle synergies analysis shows altered neural strategies in women with patellofemoral pain during walking”.

We were also asked to address additional points related with the journal requirements:

1. We followed PLOS ONE’s style requirements, including those for file naming.

2. We do not consider our author-generated code a central part of the manuscript. Key code includes the extraction of muscle synergies components using NNMF (references to the authors that first proposed this method are available in the manuscript) and SPM (open code available at www.spm1d.org). Statistical Analysis was performed with SPSS and the only MATLAB author-generated code was used to process (mostly to filter) raw data and generate figures.

3. In the resubmission, we provided the correct grant numbers in the ‘Funding information’ section: “This work was supported by the European Union’s Horizon 2020 research and innovation programme (Project EXTEND—Bidirectional Hyper-Connected Neural System) under Grant 779982 and by the Spanish MCIN/AEI/10.13039/501100011033 the “European Union NextGenerationEU/PRTR” under Grant agreement IJC2020-044467-I and by The National Council for Scientific and Technological Development – CNPq under Grant 310680/2022-0.”

4. In the previous version, we indicated that “All data are only available upon request.”. That was because data was not de-identified and there was a risk of identification of patients with PFP. In this resubmission, we indicated the following: “Data Availability: Data are available on Figshare. Normalized EMG envelopes, as well as muscle synergies components (W, H and V) from each healthy control and patient with patellofemoral pain (PFP) during walking trials at self-selected speed are available at: https://figshare.com/s/88e83948a1685ef8e956.”.

5. Regarding the ORCID number, there were some technical issues when assigning an already existing account from the corresponding author. We opted to create a new account, but the ORCID number could only be linked to one account. We asked Editorial Manager to merge the two accounts, so the problem has now been resolved.

Reviewer #1:

General commentaries:

The paper is promising, but certain issues require clarification.

My primary concern is how the complexity of neuromuscular control can differ between groups without changes in individual muscle patterns. Additionally, a more comprehensive interpretation of coordination is necessary, particularly concerning the term "muscle coordination of quadriceps" and the physiological significance of the VAF muscle.

Authors: We thank the reviewer for this comment and the suggestion. The hypothesis of muscle synergies suggests that the central nervous system (CNS) does not activate muscles individually but rather sends neural commands that activate specific muscle groups to produce the movement. In other words, muscle synergies can be thought as groups of muscles that are co-activated to produce a given movement. By activating different combinations of muscle synergies, the CNS can generate a wide range of movements using a relatively small number of control signals. A lower number of synergies indicates less complexity in terms of muscle coordination (i.e., a smaller number of independently activated synergies may hinder the proper execution of the same biomechanical subtasks of walking) and can be associated with increased co-activation levels between muscles, which can be observed in patients with neural injuries.

Variability account for (VAF) is a mathematical value used to assess the adequate reconstruction of data. In our study, VAF was used to evaluate how well a set of extracted muscle synergies can reconstruct the original muscle activation patterns (VAFtotal for the whole EMG dataset and VAFmuscle to assess the reconstruction of each individual muscle EMG). Both indices (VAFtotal and VAFmuscle) have been found to be higher in populations with neural injuries, when using a fixed number of muscle synergies to reconstruct EMG data, which indicates lower number of synergies and less complexity in terms of muscle coordination. This was also the case of our study, which suggests that women with PFP show lower motor complexity and deficit in muscle coordination to execute gait.

Regarding the concern on how complexity of neuromuscular control can differ between groups without changes in individual muscle patterns, this can be explained by at least two factors: 1) while the analysis of muscles synergies takes into account the stride-by-stride variability of each subject and group, SPM compared the mean of the linear envelopes from the whole trial, for each subject; 2) the reduced small sample size, acknowledged as a limitation in the Discussion section, may have affected the results. In fact, although qualitative differences between groups can be observed for some EMGs in Fig 2, these did not reach the significance level of 0.05. The analysis of higher sample sizes could perhaps demonstrate differences in individual muscle patterns when performing SPM analysis. In any case, when all small individual muscle differences are analyzed as a whole set of muscles with muscle synergies analysis, differences between groups become clearer. 

These points have been clarified throughout the manuscript, specifically in the Introduction and in the Discussion sections.

Specific commentaries:

Abstract:

Line 54. What joint showed differences in kinematics?

Authors: We did not perform an individual analysis on each joint to assess differences between groups, but rather used the movement deviation profile (MDP), which combines several joints and planes of motion. According to the systematic review by Gonçalves SB et al. (2022), the MDP is a more powerful approach than other statistical methods among gait analysis indices.

Gonçalves SB, Lama SBC, Silva MTD. Three decades of gait index development: A comparative review of clinical and research gait indices. Clin Biomech (Bristol, Avon). 2022 Jun;96:105682. doi: 10.1016/j.clinbiomech.2022.105682.

Introduction:

The lack of information is not always a good argument to research. Why is the PFP model relevant to study how the CNS organizes (coordinates) muscles in musculoskeletal diseases?

Authors: We appreciate and agree with the comment. PFP is a chronic pain condition characterized by a multifactorial origin. Individuals with PFP, in general, move differently from asymptomatic individuals, and it is thought that the CNS modifies motor behavior to control pain and protect the painful region by identifying a painful episode as a threat.

Moreover, studies suggest that the CNS coordinates muscle activation by modulating neural commands directed to groups of muscles combined to form muscle synergies (Sylos-Labini et al. 2017; De Groote et al. 2014). By studying muscle coordination in individuals with PFP, researchers can gain insight into how the CNS modifies muscle coordination in response to pain and how this affects movement. We included this information in the Introduction section.

Hypothesis:

The hypothesis lacks clarity. How does muscle synergy aid in comprehending the influence of a single muscle's coordination? For this reason, muscle synergy analysis may not be necessary to assess quadriceps coordination. Instead, intermuscular coordination based on time correlation could be a better alternative for this hypothesis. Please clarify.

Authors: Thank you for this comment. The muscle synergies analysis provides valuable insight about neural strategies underlying movement and functional outcomes of muscle activation. Additionally, we aimed to investigate the coordination pattern of muscles involved in segmental movement during task performance. Based on previous research and in accordance with the literature, it is known that local muscles in the region where pain is reported exhibit reduced muscle coordination. Building upon this evidence, we hypothesized that the quadriceps muscles might exhibit altered coordination in individuals with PFP. Our decision to conduct this analysis, as opposed to other approaches, stems from our specific interest in understanding the neural strategies adopted by the CNS in women with PFP. We changed this paragraph in the text.

Method.

Line: 175. Did you check if they are outliers of EMG peaks? Some authors used the median of the peaks, and others the maximum value. Please clarify

Authors: We did not perform a mathematical analysis to identify outliers. As we previously explained, “Two researchers with experience in EMG analysis and with no involvement in the data collection process guaranteed, by visual inspection, the quality of EMG data of the eight muscles for 25-30 strides per subject.”. This included the visual inspection of strides that were deemed as outliers.

Line 213. Did you check the normality of data using SPM? In the case of having a non-normal distribution, you should use the non-parametric tools of SPM. In addition, the 5% as a threshold is arbitrary. Can you give the technical reason for using this threshold?

Authors: Yes, we tested the normality of the data in the SPM code itself. As the data did not have a normal distribution, SnPM was used for all comparisons. This information has been corrected in the graphics. 

To test our null hypothesis, we calculated the critical threshold at which only 5% of smooth random curves would be expected to cross, like all classical hypotheses testing methods and the study of Robinson et al. (2015).

Robinson MA, Vanrenterghem J, Pataky TC. Statistical Parametric Mapping (SPM) for alpha-based statistical analyses of multi-muscle EMG time-series. J Electromyogr Kinesiol. 2015 Feb;25(1):14-9. doi: 10.1016/j.jelekin.2014.10.018. Epub 2014 Nov 7. PMID: 25465983.

Results:

Did you check the association between clinical measures and muscle synergy analysis, for instance, VAF values?

Authors: In a parallel analysis, which was not in the scope of this study, we tested the association between clinical measures and muscle synergy analysis but found no association.

Discussion:

Line 314. How can a decrease in muscle complexity be explained without individual muscle changes?

Authors: This question was also address in a previous comment. The decreased muscle complexity (less number of synergies) without (significant) individual muscles changes can be explained by two factors: 1) while the analysis of muscles synergies takes into account the stride-by-stride variability of each subject and group, SPM compares the mean of the linear envelopes from the whole trial, for each subject; 2) the reduced small sample size, acknowledged as a limitation in the Discussion section, may have affected the results. In fact, although qualitative differences between groups can be observed for some EMGs in Fig 2, these did not reach the significance level of 0.05. The analysis of higher sample sizes could perhaps demonstrate differences in individual muscle patterns when performing SPM analysis. In any case, when all small individual muscle differences are analyzed as a whole set of muscles with muscle synergies analysis, differences between groups become clearer.

Line 315. But muscle coordination at what level?

Authors: Results showed no significant differences between groups in terms of recruitment of each synergy throughout the gait cycle (Hs) and the weight of each muscle in each synergy (Ws) (Fig 3). Therefore, we cannot speculate at which level did differences in muscle coordination happen (e.g., spinal cord, brainstem or and cerebral cortex). However, we did find significant differences in terms of the number of muscle synergies and VAF indices, which demonstrates overall differences in muscle coordination between groups.

Reviewer #2: This study investigated muscle activation and lower kinematics differences between PFP and normal counterparts. The results revealed that women with PFP show lower motor complexity and deficit in muscle coordination to execute gait, indicating that gait in PFP gait is the result of different neural commands compared to asymptomatic women. The title is not a direct connection between muscle synergy and neural commands. Besides, this study doesn’t prove neural commands changed. The same issue exists in the lines 45-47.

Authors: The muscle synergy analysis allows us to gain a better understanding of the neural strategies underlying muscle coordination. This analysis provides insight into how the CNS commands motor patterns to execute the movement. We appreciate your comment and have made the change from “neural command” to “neural strategies” in the text and title. 

Lines 65-68, a more comprehensive background regarding reasons causing PFP, treatment procedure, and biomechanics change is expected.

Authors: Thank you for this comment. We have revised this part of the paragraph; however, we have chosen not to delve into the specific biomechanical alteration. The literature extensively covers these changes, and our objective was to concentrate on contextualizing the neural strategies employed to control the motor patterns observed during walking.

The reason why only female subjects were selected is not explained in the manuscript. I would not feel stranger for adding this as one of the limitations.

Authors: The fact that we only recruited women with PFP to participate in this study has been included as a limitation: “Finally, this study was carried out only with women volunteers, as women are twice as likely as men are to develop patellofemoral pain. Although muscle synergies are similar in younger adults between females and males, this similarity is lower in older females as compared to older males. Therefore, this study should be replicated in men volunteers presenting PFP to assess differences in motor control during gait compared to men with no PFP.”.

Lines 134-137, The authors took space to introduce experimental settings, but the test protocol was too ambiguous.

Authors: Regarding the part of text that says “All participants walked at self-selected speed on a 6-meter-long by 1-meter-wide track.”, we think that this is not ambiguous. If the reviewer feels that “Data from at least 25-30 strides were collected, where corresponding EMG signals presented no artifacts or noise that would prevent a trustful analysis.”, we then explained that “Two researchers with experience in EMG analysis and with no involvement in the data collection process guaranteed, by visual inspection, the quality of EMG data of the eight muscles for 25-30 strides per subject.”.

Lines 154-155, please provide more details for identifying gait spatiotemporal parameters.

Authors: We thank the reviewer for this observation. Indeed, it was necessary to provide more details, which have been included in the revised version: “Heel strike events from each walking trial were determined by visual inspection after analyzing Vicon data, and were used to define the beginning and the end of each stride”.

In Table 1, SF-36, some indexes were confusing. Adding related content in the text.

Authors: We thank the reviewers for this observation. SF-36 is a questionnaire that assesses quality of life and is divided into eight domains, each evaluating a specific aspect of quality of life and assigned an individual score. We have reorganized the order of the domains in the table to enhance reader comprehension and have made corresponding adjustments to the text.

---

## [Decision Letter · Decision Letter 1]

17 Jul 2023

PONE-D-23-03138R1MUSCLE SYNERGIES ANALYSIS SHOWS ALTERED NEURAL STRATEGIES IN WOMEN WITH PATELLOFEMORAL PAIN DURING WALKINGPLOS ONE

Dear Dr. Oliveira Barroso,

Thank you for submitting your manuscript to PLOS ONE. After careful consideration, we feel that it has merit but does not fully meet PLOS ONE’s publication criteria as it currently stands. Therefore, we invite you to submit a revised version of the manuscript that addresses the points raised during the review process.

We look forward to receiving your revised manuscript.

Kind regards,

Yaodong Gu

Academic Editor

PLOS ONE

Journal Requirements:

Additional Editor Comments (if provided):

Please check these minor questions raised by the reviewers.

Reviewers' comments:

Reviewer's Responses to Questions

**Comments to the Author**

1. If the authors have adequately addressed your comments raised in a previous round of review and you feel that this manuscript is now acceptable for publication, you may indicate that here to bypass the “Comments to the Author” section, enter your conflict of interest statement in the “Confidential to Editor” section, and submit your "Accept" recommendation.

Reviewer #1: (No Response)

Reviewer #2: (No Response)

2. Is the manuscript technically sound, and do the data support the conclusions?

Reviewer #1: Yes

Reviewer #2: Yes

3. Has the statistical analysis been performed appropriately and rigorously? 

Reviewer #1: Yes

Reviewer #2: Yes

4. Have the authors made all data underlying the findings in their manuscript fully available?

Reviewer #1: Yes

Reviewer #2: Yes

5. Is the manuscript presented in an intelligible fashion and written in standard English?

Reviewer #1: Yes

Reviewer #2: Yes

6. Review Comments to the Author

Reviewer #1: Thank you to the authors for responding to my questions. However, in my opinion, two key points have not been resolved in this revision.

The lack of differences at individual muscle levels needs to be better discussed in the paper. In studies examining neurological diseases (e.g., cerebral palsy or stroke), significant differences have been reported in the number of synergies, VAF, and the pattern of individual muscles. My concern is to give support with example of other studies published where the individual muscle pattern showed no differences, while the muscle synergy analysis did find.

My second concern is about the lack of association between muscle synergy and clinical variables. Even though this was not the aim of the study, I believe discussing the lack of associations could help suggest that neural strategies might not correlate with clinical symptoms such as pain level or psychological variables (e.g fear of movement, catastrophizing, or depression).

Overall, the revised version may be improved if the authors give some space to discuss the above points and contrast the results with the existing literature.

Reviewer #2: Lines 185-187, the author shall demonstrate the reason for using MDP in this study rather than arbitrarily sum up that it is a better statistical method than others.

7. PLOS authors have the option to publish the peer review history of their article (what does this mean?). If published, this will include your full peer review and any attached files.

Reviewer #1: No

Reviewer #2: No

---

## [Author Response · Author response to Decision Letter 1]

11 Sep 2023

Comments from Editor and reviewers:

Editor Comments:

Please check these minor questions raised by the reviewers.

Authors: We thank the Editor for this comment. All minor questions raised by the reviewers are carefully addressed below. Changes related with each of these questions are highlighted in blue in the revised version of the manuscript.

Reviewer #1:

Thank you to the authors for responding to my questions. However, in my opinion, two key points have not been resolved in this revision.

The lack of differences at individual muscle levels needs to be better discussed in the paper. In studies examining neurological diseases (e.g., cerebral palsy or stroke), significant differences have been reported in the number of synergies, VAF, and the pattern of individual muscles. My concern is to give support with example of other studies published where the individual muscle pattern showed no differences, while the muscle synergy analysis did find.

Authors: We thank the reviewer for raising this point again. In the previous revision, we did not understand that the reviewer suggested that if differences in muscle synergies were reported in a given neurological disease, individual muscle EMG differences should (in theory) be also reported.

In addition to the analysis of muscle synergies, differences in individual EMG patterns have been found in some neurological diseases such as stroke (Srivastava et al, 2019; Forman et al, 2019) and cerebral palsy (Cappellini et al, 2023; Forman et al, 2019; Michelsen et al, 2020), for example, but also in PFP (Miao et al, 2015; Bley et al, 2014; Kalytczak et al, 2018). However, these studies mostly focused on hip and knee muscles and compared differences in the activation onset time between pairs of muscles. In our study, we used SPM analysis to look for individual differences in muscle activation between PFPG and CG, but did not find significant differences. In this regard, it is important to highlight that while SPM compares the mean of the linear envelopes of the EMG signals from each subject of each group, the analysis of muscles synergies considers the stride-by-stride variability of each subject and group. Moreover, although we did find differences in the number of muscle synergies between the two groups, no differences were found in terms of recruitment of each synergy throughout the gait cycle (Hs) and the weight of each muscle in each synergy (Ws).

The reduced small sample size, acknowledged as a limitation in the Discussion section, may have affected the results. In fact, although qualitative differences between groups can be observed for some EMGs in Fig 2, these did not reach the significance level of 0.05. The analysis of higher sample sizes could perhaps demonstrate differences in individual muscle patterns when performing SPM analysis.

In any case, when all small individual muscle differences are analyzed as a whole set of muscles with muscle synergies analysis, differences between groups become clearer. This has been explained in lines 393-406 and 443-448 of the revised version of the manuscript.

My second concern is about the lack of association between muscle synergy and clinical variables. Even though this was not the aim of the study, I believe discussing the lack of associations could help suggest that neural strategies might not correlate with clinical symptoms such as pain level or psychological variables (e.g fear of movement, catastrophizing, or depression). Overall, the revised version may be improved if the authors give some space to discuss the above points and contrast the results with the existing literature.

Authors: As suggested by the reviewer, we have included a paragraph discussing the lack of associations between neural strategies and clinical symptoms (lines 413-434).

It is plausible that clinical and psychosocial variables such as catastrophizing, fear of movement and pain would lead individuals with PFP to use different neural strategies to protect against pain during movement. Studies showed an association between kinesiophobia and the reduction of cadence and peak of knee flexion during stair descent (Silva et al, 2019), as well as with peak of internal hip rotation increase during single-leg drop vertical jump (Vasconcelos et al, 2023). In a parallel analysis, which was not in the scope of this study, we found no association between VAFtotal and clinical variables of pain intensity (NRPS), disability (AKPS), fear of movement (TAMPA), catastrophizing (PCS) and depression (Beck scale). In individuals who suffered an incomplete spinal cord injury, no association was also observed between the number of synergies and the functional scales used to evaluate these patients, indicating that neural strategies for movement performance do not necessarily relate to clinical variables (Pérez-Nombela et al, 2017). Individuals with PFP present high levels of catastrophizing, fear of movement, depression, and anxiety and these variables are associated with pain and disability (Maclachlan et al, 2017). However, catastrophizing and kinesiophobia are not associated with objective function measured in tests such as forward step-down, single leg hop test, and modified star balance test (Priore et al, 2019). In this regard, it is important to highlight that the relationship between biomechanical and clinical variables is still unclear. Therefore, there is a need for future studies with a larger sample size that may better clarify the relationship between neural strategies and clinical and psychosocial variables, which can help achieve better clinical management of patients with PFP.

Reviewer #2: 

Lines 185-187, the author shall demonstrate the reason for using MDP in this study rather than arbitrarily sum up that it is a better statistical method than others.

Authors: MDP was chosen for unifying and simplifying the understanding of kinematic data by combining all analyzed data into a single curve and calculating the deviation of the movement of the PFPG in relation to the CG throughout the movement cycle (Barton et al, 2012). MDP was applied in previous studies by our group to compare the kinematics of women with PFP with the kinematics of asymptomatic women in various tasks, including gait, proving to be a suitable method for analyzing possible kinematic differences in this population (Lopes Ferreira et al, 2019 and 2020). In addition, the analysis of kinematics through gait indices is a common practice, and among the indices used, the MDP is a more powerful approach than other statistical methods, according to the systematic review presented by Gonçalves SB et al. (2022). We included this information in the text (lines 185-195).

---

## [Decision Letter · Decision Letter 2]

22 Sep 2023

MUSCLE SYNERGIES ANALYSIS SHOWS ALTERED NEURAL STRATEGIES IN WOMEN WITH PATELLOFEMORAL PAIN DURING WALKING

PONE-D-23-03138R2

Dear Dr. Oliveira Barroso,

We’re pleased to inform you that your manuscript has been judged scientifically suitable for publication and will be formally accepted for publication once it meets all outstanding technical requirements.

Kind regards,

Yaodong Gu

Academic Editor

PLOS ONE

Additional Editor Comments (optional):

Well done!

Reviewers' comments:

Reviewer's Responses to Questions

**Comments to the Author**

1. If the authors have adequately addressed your comments raised in a previous round of review and you feel that this manuscript is now acceptable for publication, you may indicate that here to bypass the “Comments to the Author” section, enter your conflict of interest statement in the “Confidential to Editor” section, and submit your "Accept" recommendation.

Reviewer #1: All comments have been addressed

Reviewer #2: All comments have been addressed

2. Is the manuscript technically sound, and do the data support the conclusions?

Reviewer #1: Yes

Reviewer #2: Yes

3. Has the statistical analysis been performed appropriately and rigorously? 

Reviewer #1: Yes

Reviewer #2: Yes

4. Have the authors made all data underlying the findings in their manuscript fully available?

Reviewer #1: Yes

Reviewer #2: Yes

5. Is the manuscript presented in an intelligible fashion and written in standard English?

Reviewer #1: Yes

Reviewer #2: Yes

6. Review Comments to the Author

Reviewer #1: The authors have addressed all of the suggestions and comments I provided. Success with the publication.

Reviewer #2: (No Response)

7. PLOS authors have the option to publish the peer review history of their article (what does this mean?). If published, this will include your full peer review and any attached files.

Reviewer #1: No

Reviewer #2: No

---

## [Editor Report · Acceptance letter]

26 Sep 2023

PONE-D-23-03138R2 

MUSCLE SYNERGIES ANALYSIS SHOWS ALTERED NEURAL STRATEGIES IN WOMEN WITH PATELLOFEMORAL PAIN DURING WALKING 

Dear Dr. Oliveira Barroso:

I'm pleased to inform you that your manuscript has been deemed suitable for publication in PLOS ONE. Congratulations! Your manuscript is now with our production department. 

Kind regards, 

on behalf of

Professor Yaodong Gu 

Academic Editor

PLOS ONE